# Willingness, motivators and barriers to time bank participation: A scoping review

Nur Shafira Azizan[1], Chong Chin Che[1]*, Mei Chan Chong[1], Maw Pin Tan[2], Yong Kek Pang[2,a,‡], Khadijah Alavi[3,b,‡]

1 Department of Nursing Science, Faculty of Medicine, Universiti Malaya, Malaysia, 2 Department of Medicine, Faculty of Medicine, Universiti Malaya, Malaysia, 3 Centre of Study for Psychology & Human Well-Being, National University of Malaysia, Malaysia

☯ These authors contributed equally to this work
‡ These authors also contributed equally to this work
¤a Current Address: Department of Nursing Science, Faculty of Medicine, Universiti Malaya, Kuala Lumpur, Malaysia
¤b Current Address: Research Center for Psychology and Human Wellbeing, Faculty of Social Sciences and Humanities, 34600 Universiti Kebangsaan Malaysia, Bangi
* chechongchin@um.edu.my

## Abstract

This scoping review systematically identifies the extent and nature of evidence regarding willingness, motivators, and barriers influencing community participation in time bank systems, with a focus on older adults. Time bank is recognized as an effective tool for enhancing social cohesion and reducing isolation, particularly among older adults. However, specific factors such as barriers and motivators driving participation remain underexplored. This review adhered to Arksey and O'Malley's framework and PRISMA-ScR guidelines. A systematic search of PubMed, Embase, CINAHL, Web of Science, Cochrane Library, Google Scholar, and Science Direct (January 2010–December 2024) identified studies on willingness, motivators, and barriers to time bank participation. Two reviewers independently screened titles, abstracts, and full texts, extracting data using a predefined tool. Thirty studies (quantitative, qualitative, and mixed-methods) were included. Findings revealed high willingness among younger populations, particularly university students, driven by intrinsic motivations like altruism and meaningful engagement. Healthcare professionals and older adults showed moderate willingness, influenced by health needs and caregiving roles. Key motivators included robust willingness to participate across different population, social and community influences, demographic and structural determinants, institutional and organizational support. Barriers included psychological and perceptual barriers, structural and systemic barriers, technological and usability, social and community, economic and policy-related barriers. Time bank holds significant potential for fostering community engagement, but challenges such as usability and trust must be addressed. Tailored strategies, improved platform design, and supportive policies are essential to maximize participation. Future research should

**Data availability statement:** This study is a scoping review. All data consist of information extracted from published sources, which are fully cited in the paper. The charted data used in the review (study characteristics and extracted variables) are provided in Supporting Information. No additional raw data were generated.

**Funding:** We want to thank the Ministry of Higher Education for the Fundamental Research Grant Scheme (FRGS/1/2023/SS10/UM/02/5) awarded to Dr Che Chong Chin.

**Competing interests:** The authors have declared that no competing interests exist.

address identified gaps to develop inclusive, scalable, and sustainable time bank systems that meet diverse community needs.

---

# 1 Introduction

The concept of time bank, a system of reciprocal exchange where services are traded using units of time rather than monetary currency, has gained global attention as a sustainable and community-driven approach to address various social and economic challenges [1–4]. Time banks foster community cohesion, promote interdependence, and encourage active participation among members, making them particularly relevant in aging societies [5–7]. Older adults, a demographic often facing social isolation, financial constraints, and health-related challenges, stand to benefit significantly from time bank initiatives [8–11].

Older adults represent a vital and expanding part of communities, bringing with them a wealth of skills, knowledge, and lived experiences. Their participation in social initiatives such as time banking enables meaningful intergenerational connections and contributes to stronger, more cohesive communities [12–14]. Time banks provide a platform for older adults not only to access assistance but also to share their abilities and wisdom, reinforcing a sense of purpose, belonging, and engagement [11,15,16]. This approach aligns with global movements toward active and healthy aging, which emphasize opportunities for continued contribution and participation throughout later life [8,17,18].

Research shows that older adults' engagement with time banking is shaped by a balance of motivators and obstacles. Opportunities for social interaction, reciprocity, altruism, and personal fulfillment encourage participation [19–23], while barriers such as limited awareness, low digital literacy, and concerns regarding trust and safety may constrain involvement [7,24–27]. These findings, however, are scattered across diverse disciplines including gerontology, public health, and social work, and are presented through varied methodological approaches ranging from small-scale qualitative studies to large-scale surveys [27–29].

Although reviews have examined time banking more broadly as a model of social innovation [1–3,6,30], no comprehensive synthesis has focused specifically on older adults. The lack of an integrated review limits understanding of how older adults perceive and participate in these systems, and hinders the development of tailored strategies that reflect their strengths and preferences.

This scoping review seeks to address that gap by systematically mapping existing evidence on older adults' engagement with time banking. It examines three interconnected dimensions: willingness to participate, obstacles that may hinder involvement, and motivators that sustain engagement. Through this synthesis, the review aims to provide a clearer understanding of how time banks can serve as inclusive platforms that recognize and support the contributions of older adults, while also enhancing their well-being and social connectedness. The findings are intended to inform policy, guide practice, and inspire further research into strengthening community-based exchange systems that value the participation of people at every stage of life.

## 2 Methods

### 2.1 Protocol and registration

Arksey and O'Malley's framework [31] for scoping reviews was adopted as the foundation as well as an updated framework by the Joanna Briggs Institute. According to this framework, there are five different stages which include; identifying the research question, identifying relevant studies, selecting studies, charting the data, collating, summarising and reporting results. The scoping review also adhered to the Preferred Reporting Items for Systematic Reviews and Meta-Analyses Extension for Scoping Reviews (PRISMA-ScR) [32].

The protocol have been registered in the Open Science Framework (Registration DOI 10.17605/OSF.IO/X9KA4) in March 2025. All extracted data were checked for accuracy by one author (CCC) who did not partake in the data extraction. As per JBI Scoping Review guidance, we did not conduct a quality assessment.

### 2.2 Identifying the research question

The formulation of the research question and sub-questions was grounded in a preliminary review of existing literature on time banking [31–34]. This preliminary mapping revealed three recurring themes. First, several studies have highlighted the community's willingness to participate in time bank systems, particularly among older adults [9,11,33], caregivers [15], and students [13,27,28]. Second, other studies have emphasized motivational factors such as reciprocity, altruism, social value, and personal development as key drivers for participation [12,19–23]. Third, the literature also documents barriers such as lack of awareness, social stigma, organizational challenges, and trust or safety concerns [2,7,10,12,24,25,29]. Drawing directly on these established themes, this scoping review was designed to answer the overarching research question: What is known about the perspectives in participation of a time bank system among communities? Specifically, the review addressed three sub-questions::

1. What is the community level of willingness in participation of a time bank system?

2. What motivates the community to participate in a time bank system?

3. What are the community existing barriers in participation of a time bank system?

### 2.3 Identifying relevant studies

A comprehensive search to identify studies on the willingness, perceived barriers and motivators in participation in time bank system for older adults from January 2010 to December 2024 was selected to ensure that the review captured both the early and more recent developments of time bank research. Empirical studies on time banking began to increase noticeably after 2010, coinciding with the growth of digital platforms and community-based exchange models that made the system more widely accessible and applicable. Earlier publications (prior to 2010) were limited, often conceptual or descriptive in nature, and less relevant to current models of participation. The inclusion of studies up to December 2024 ensured that the review reflects the most up-to-date evidence available at the time of study.

In addition to electronic databases (MEDLINE, Embase, CINAHL, Cochrane Library, ScienceDirect), supplementary searches were conducted in Google Scholar and relevant grey literature sources (Grey Literature Report, OpenGrey, Web of Science Conference Proceedings, government documents, and academic theses/dissertations). Although Google Scholar and grey literature sources were included to capture potentially relevant evidence beyond peer-reviewed databases, all records from these sources underwent the same predefined inclusion and exclusion screening as other studies. Recognizing the variability in quality, a multi-step screening process was applied to ensure credibility. Consistent with scoping review methodology, no formal critical appraisal was conducted, as the primary aim was to map the breadth of available evidence. Nonetheless, studies that lacked methodological clarity or relevance were excluded, and only those

with sufficient transparency and applicability were retained. This strategy reduced the risk of incorporating low-quality or non-credible sources while maintaining comprehensive coverage and minimizing publication bias.

For this review, the electronic search strategy was systematically developed to ensure replicability and comprehensiveness. In MEDLINE (via PubMed), for example, the search combined both free-text keywords and Medical Subject Headings (MeSH). Because the concept of time banking is described in diverse ways, the strategy included not only direct terms such as timebank, time bank, and timebanking, but also related expressions such as time currency, time dollar, time exchange, service exchange, peer-to-peer exchange, and community exchange. Boolean operators (AND/OR) and truncation symbols (e.g., timebank*, volunteer*) were applied to capture variations in spelling and word endings. An example of the PubMed search string (("Time Banking"[tiab] OR "Time Bank*"[tiab] OR timebank*[tiab] OR "Time Currency"[tiab] OR "Time Dollar*"[tiab] OR "Time Exchange*"[tiab] OR "Service Exchange*"[tiab] OR "Community Exchange*"[tiab] OR "Peer-to-Peer Exchange*"[tiab])) AND ("Community Participation"[MeSH] OR "Social Participation"[MeSH] OR "Volunteers"[MeSH] OR "Community-Institutional Relations"[MeSH] OR "Intergenerational Relations"[MeSH] OR "community participation"[tiab] OR "social engagement"[tiab] OR volunteer*[tiab] OR "intergenerational support"[tiab]). Applied filters: English; Publication date from 2010/01/01–2024/12/31 that gained 13 results.

## 2.4 Study selection

Preferred Reporting Items for Systematic Reviews and Meta-Analyses Extension for Scoping Reviews (PRISMA-ScR) was used in the selection process of the study. At first, two reviewers (NSA and CCC) screened the titles and abstracts of the articles. During this process, the following steps were implemented: 1) if both reviewers concurred on the inclusion of an article, it was subsequently subjected to a full-text review by each reviewer; 2) if both reviewers agreed on the exclusion of an article, it was removed from the study; 3) if there was a lack of consensus between the reviewers regarding an article's inclusion or exclusion, it advanced to the next phase of the screening process, where each reviewer conducted a full-text assessment before reaching a final decision. In this phase, both reviewers independently performed a comprehensive review of the selected articles. The inclusion and exclusion criteria were established in alignment with the research objectives.

**Inclusion criteria.** Data from the articles are from qualitative, quantitative, and mixed methods forms of the original studies. The studies eligible for inclusion in the review had to focus on the participants willingness, motivators and barriers towards time bank participation. The research involved all layers of the population, the time bank system, and associated data. The articles were published in the English language. ***Exclusion criteria:*** Studies focusing on populations not relevant to time bank system (e.g., corporate barter systems, traditional volunteering unrelated to time bank); studies involving individuals or communities where time bank is not applicable or feasible (e.g., isolated populations without access to a time bank system); studies addressing economic systems or currency models that are unrelated to time bank (e.g., blockchain or cryptocurrency); theoretical papers without empirical data or practical insights into participation in time bank; reviews or meta-analyses that do not provide original data and studies using methods that are unsuitable for addressing willingness, barriers, or motivators (e.g., engineering-focused system design studies). Such studies would not contribute to answering the research questions.

## 2.5 Charting the data

The research team collaboratively develops a data-charting form using Microsoft Excel version 16.16.27 to gather and tabulate all relevant data from the studies that have been selected to identify key variables for extraction that will contribute to addressing the research question. Data were extracted on participant characteristics (e.g., age, gender, socioeconomic status), study context and setting, details of the time bank intervention or exposure, outcomes related to willingness, motivators, and barriers to participation, study design, sample size, follow-up duration, and data collection methods. To enable synthesis, several assumptions and simplifications were applied: demographic and outcome variables were treated as comparable across studies despite minor differences in reporting; continuous variables were grouped into broader categories;

missing data were coded as "not reported"; all studies were included regardless of quality with methodological limitations noted; and different measures of the same construct were considered equivalent with the instrument recorded.

## 2.6 Collating, summarising and reporting the results

The fundamental characteristics of the included studies were systematically analyzed using qualitative thematic analysis and organized in a tabular format. The main characteristic of the study such as author's name, publication year, country of the study conducted, study design, study aims, sample size, and key findings were summarized. S1 Table summarizes the outcomes of this synthesis. Following the collating of article details and content, the researchers (NSA & CCC) developed the themes and domains, subsequently reviewed by the research team. Based on this classification, key domains were established to encapsulate essential aspects of the published research on willingness to participate in time bank system, as well as the motivators and barriers influencing their participation. The team identified gaps in our understanding of the current state of research. The discussion is structured based on the themes that emerge.

# 3 Results

## 3.1 Search outcomes

The searches for primary studies and grey literature located 1120 items were relevant hits to from these databases using the keywords/phrase time bank, time bank, time bank, time-bank and time-bank. As this topic has long been discussed by many researchers, only articles that most pertained to research questions have been selected by inserting keywords such as "motivation", "barriers" and "willingness". Limiting the literature yield to those which were from peer-reviewed journals, those which are relevant to community, caring and healthcare, those were in the English Language, and those which have full-text availability online, the yield was lowered to 233. The next screening was sifting the abstract of the articles, which was decided by reviewing the relevance of the article with the topic. The abstract sifting produced 42 findings. The remaining articles were then examined using full text filtering, and their relevance to and interest in the study's subject matter were evaluated. After applying inclusion and exclusion criteria, the search's final output was reduced to 30 articles. These 30 articles are relevant for the factors influencing, motivation, barriers and willingness of participation time bank. The PRISMA flowchart of the study selection process was generated using PRISMA Flow Diagram tool [32] (S1 Figure).

## 3.2 Characteristics of included studies

The included studies comprised: 24 articles peer-reviewed publications; six grey literatures. Ten quantitative studies, 15 qualitative studies and five mixed-method studies are relevant to the willingness, motivator and barriers of participation in time bank system. Although the populations and the objectives of the included studies were different, it was possible to categorize the results in 1) willingness to participate in time bank system, 2) motivators to participate and 3) barriers to participate. Among 30 studies, eight were in China, five in the United States, two were in the United Kingdom, two in Spain and one study from Thailand, Nigeria, Sweden, India, New Zealand, Russia, Turkey, Hong Kong, England and Belgium, one study combined population from Bulgaria, Denmark, England, France, Spain, one study combined population from United States, New Zealand and India, and one study combined population from Finland and United Kingdom (S1 Table and S2 Figure).

## 3.3 Synthesis of result

### 3.3.1 Willingness to participate in time bank system. *Robust willingness to participate across different population*

Across studies, a majority of participants express interest in time bank system, with variations by age, gender, and socioeconomic factors. University students and younger generations demonstrate the highest levels of willingness, with

82.67% of Chinese university students [27] and 71% of Thailand youth aged 18–23 [28] expressing interest in participating. Similarly, 62.7% of respondents from the general community in Hangzhou reported being "very willing" or "willing" to engage in time bank services, with gender, age, and education level playing significant roles [17].

Healthcare professionals, such as nurses, also show a moderate willingness, with 59.6% expressing interest in volunteering for older adults with disabilities, influenced by positive attitudes and reduced barriers [18]. Among older adults, willingness varies. 58% of Chinese older adults expressed readiness to engage in healthcare-related time bank, driven by physical quality of life and demographic factors like gender and residence [8]. Meanwhile, rural elderly populations demonstrate a relatively higher demand for mutual support services [9].

Caregivers of dementia patients exhibited higher willingness when factors like caregiver burden, technology affinity, and openness to professional support were present [15]. These findings underscore that willingness is often shaped by specific enablers such as personal attitudes, social environments, perceived value, and ease of participation.

### Factors influence the willingness to participate in time bank system

Physical quality of life plays a significant role, with older adults who have better physical well-being showing greater enthusiasm for participation [8]. In contrast, individuals experiencing lower mental well-being tend to be less willing to engage in time bank, suggesting that mental health challenges may serve as a barrier [8]. Social and economic factors also shape attitudes toward time bank, as value judgment, social support, and overall socioeconomic conditions determine the level of engagement [27]. Additionally, caregiver burden has been identified as a driving factor, with those experiencing higher caregiving responsibilities showing a stronger inclination to participate in time bank [15].

### Age and regional differences further influence participation willingness

Studies indicate that younger individuals, particularly those between the ages of 18 and 23, demonstrate the highest willingness to engage in time bank, with 71% expressing interest [28]. In contrast, for elderly individuals living in rural areas, participation is largely determined by perceived usefulness, behavioral attitudes, and subjective norms [10]. These findings suggest that while time bank holds broad appeal, the specific motivations and barriers vary across age groups and geographic locations. For example, for the older people group, the free time every day has different effects on the willingness to participate in online timebank. Three to five hours of free time has a significantly positive effect on the willingness to participate in online timebank in the older people group. However, free time of six hours or more has a significantly negative effect on the willingness to participate in online timebank in the older people group. As a result, older individuals are less likely to join in online timebanks because they are too busy caring for their grandkids or pursuing their own interests [16].

### 3.3.1 Motivators to participate in time bank system. *Personal and psychological motivations*

The motivations behind participation in time bank systems can be examined through several interrelated themes, drawing from established theoretical frameworks and empirical studies. Personal and psychological motivations play a significant role in influencing engagement. In this review, studies have shown that participants in time banks often seek personal growth, self-promotion, and the opportunity to contribute meaningfully to society [6,11,12,19,21–23,33]. This is aligned with Self-Determination Theory [19], individuals are intrinsically motivated to participate in activities that enhance their sense of self-fulfillment and meaningful engagement. Additionally, the concept of altruism and social contribution that rooted in the theory of reciprocal altruism [20], explains why many individuals participate in time bank with the belief that fairness and reciprocity underpin social exchanges as reported by [21] and [23]. These motivations align with the idea that individuals engage in prosocial behaviors when they perceive mutual benefits, reinforcing trust within the community. Moreover, it was also reported that perceived value and rewards, influence participation through factors such as ease of use, perceived usefulness, and tangible benefits like time credits [3,4,12,22–24,29]. This suggests that when participants find the system user-friendly and beneficial, they are more likely to engage in sustained participation.

### Social and community influences

This scoping review also found that social and community influences significantly shape engagement in time bank. For instance, older adults and caregivers, in particular, rely on social support systems that time banks provide, allowing them to connect with peers and access needed services [5,6,9–11,13,33]. This is aligned with social capital theory [35]highlights the importance of social networks and peer support in fostering a sense of belonging and engagement. Furthermore, review also found that trust and reciprocity, which are fundamental principles of social exchange theory [36], enhance long-term commitment by fostering a system of mutual aid where participants expect their contributions to be reciprocated [23]. Additionally, cultural and societal norms influence participation, particularly in communities where collectivist values are prevalent. It was evident that participation in time banks is often guided by societal expectations and local customs, which play a more prominent role in rural settings where social cohesion and communal participation are strong motivating factors [17,27].

### Demographic and structural determinants

Demographic and structural determinants further influence engagement, demonstrating the role of age, education, health, and economic conditions. Studies have shown that younger individuals, particularly urban dwellers, exhibit higher engagement due to their greater digital literacy and social capital, while older adults tend to participate when specific conditions, such as health benefits or caregiving responsibilities, make time bank more relevant to their needs [8,16]. What's more, education and experience also shape participation, as higher levels of education and prior volunteer experience are positively correlated with increased involvement in time banks, which posits that investment in education enhances social and economic engagement [10,16]. Studies also reported that health and economic status influence participation patterns, with elderly individuals in good health being more actively engaged, while socioeconomic conditions determine the extent to which individuals rely on time banks as a support mechanism [10,16].

### Institutional and organizational support

Institutional and organizational support is crucial in sustaining participation in time bank systems. Studies reported that well-defined program structures, strong management, and clear guidelines enhance participation by providing reliability and consistency in exchanges [11,12]. Moreover, engagement strategies rooted in motivation theories, such as Herzberg's Two-Factor Theory [37] suggest that both intrinsic motivators (recognition and meaningful work) and extrinsic incentives (rewards and structured meetings) improve retention. This review found that regular meetings and well-planned incentive structures foster sustained participation by reinforcing a sense of accountability and community [11,12]. These factors collectively highlight the need for comprehensive structural support to maintain the long-term viability of time bank initiatives.

### 3.3.2 Barriers to participate in time bank system. *Psychological and perceptual barriers*

This review found that concerns about mental quality of life and the reluctance to transition from traditional market-driven exchanges to community-based reciprocity [6,8]. This is align with the Theory of Planned Behavior (TPB) [38]. According to TPB, individuals' behavioral intentions are influenced by attitudes, subjective norms, and perceived behavioral control. Furthermore, participants who perceive time bank as complicated or impractical are less likely to participate, reinforcing the need for user-friendly and transparent platforms. For instance, [29] reported that the negative perceptions of time bank, including concerns about its usefulness and ease of use suggest that participants may hold unfavorable attitudes toward its adoption, thereby reducing engagement. Additionally, the lack of awareness and trust in time bank systems reported by [25] emphasizes the role of perceived usefulness and ease of use in influencing technology adoption.

### Structural and systemic barriers

Literature analysis revealed that institutional constraints, including regulatory frameworks, organizational inefficiencies, and resource limitations, shape individual participation in structured systems like time bank. For instance, when time bank

institutions lack clear policies, participants struggle to navigate the system effectively, leading to disengagement [2,7]. The inefficiencies in resource allocation and operational challenges in credit systems were also identified as structural and systemic challenges [2,7]. For instance, challenges such as time credit hoarding and accessibility issues. It was reported that time banks generally experience difficulties explaining to prospective members precisely what they might be used for and how they operate [2,3,7,13]. Since particular skills can be in short supply, there is no guarantee that a time bank will meet everyone's demands. This will then posed potential for a mismatch between the skills individuals offer and the skills they need. Participants may find themselves in situations where they have an abundance of one skill but require assistance in a different area which resulting in time credit hoarding and accessibility issues [2,3,7,13]. This suggest that if time bank fails to offer tangible benefits or requires excessive effort, individuals may opt out due to an imbalanced cost-benefit analysis.

### Technological and usability challenges

Technological and usability challenges further compound these structural issues, particularly when platforms are perceived as un-intuitive or difficult to navigate, lack user-friendly interfaces or adequate onboarding processes. Studies suggest that when users perceive a time bank platform as easy to use, they tend to have a positive attitude toward both making requests for help and offering services [6,29]. However, when users see the platform as useful, their attitudes toward both actions become more negative, which may indicate hesitation or other concerns despite recognizing its benefits [6,29]. Additionally, while having a positive attitude toward making requests increases the likelihood of actually requesting help, the same does not apply to offering services [6,29]. This means that even if users feel good about providing help, they may not necessarily follow through with offering their services and potential participants may perceive them as overly complex, reducing their willingness to engage. This indicates the importance of platform design in fostering ease of use and accessibility to encourage broader participation.

### Social and community constraints

Evidence from included literatures also found that the lack of peer support and limited recognition of diverse skills hinders participation of time bank system [13,21,26]. This suggests that when individuals do not receive encouragement or support from their peers, they may feel hesitant or unmotivated to engage in the system. This lack of a supportive community can reduce trust and willingness to exchange services. Additionally, if the platform does not adequately recognize and value a wide range of skills, potential participants may feel that their abilities are not appreciated or relevant, leading to disengagement [13,26]. When participants do not perceive strong social bonds or mutual trust within the community, lack of socialization among participants their likelihood of sustained engagement diminishes [2,21]. As a result, these challenges limit overall participation and the effectiveness of the time bank system. Disparities in skill recognition and unequal participation opportunities reinforce issues of social exclusion and stratification, limiting time bank's inclusivity and effectiveness.

### Economic and policy-related barriers

Time bank organizations and initiatives depend on external resources, such as funding, public space access, socially impactful government policies, institutional backing, and the transition to digital services are examples of external obstacles to sustain their operations [14,30]. Studies reported that the lack of financial incentives, funding shortages, and the absence of government support posed additional difficulties for the time bank system [30]. What's more, time banks have internal obstacles such as members' differences over how to redefine labour, exchange rates, and the largely homogeneous membership population [14]. The absence policy on these discouraged engagement in time bank system.

Additionally, this review found that the integration of time bank within formal economic structures remains challenging as the system operates outside traditional monetary frameworks, creating tensions between informal and institutionalized economic practices. Integrating time bank into the broader economic framework presents challenges related to taxation and regulation, as it blurs the distinction between voluntary work and taxable employment. In support, a qualitative study

that compares Finland and UK time bank found that in Finland, services exchanged through timebanks should be reported to the tax authority in their euro value, as the timebank generates economic benefits to its members an tax liability had hinders the community to actively engage in time bank [30].

In contrary, in country where taxation appears to demonstrate not a significant concern to timebanks at any point, time bank might be considered as creating market disturbances [4]. For instance, in United Kingdom the management of unemployment has been stretched quite a distance and timebanks have encountered pressure from job centres [30]. As an example, an unemployed professional painter may leverage their skills to complete a house painting project in exchange for time credits. However, this practice could lead to dissatisfaction among other painters seeking employment, as it introduces a labor exchange that operates below conventional market rates.

## 4  Discussion

Time bank, a system of reciprocal service exchange based on the principle of "equal time, equal value," has gained attention as a community-oriented model fostering social support and engagement. However, participation in time bank system is influenced by a complex interplay of willingness, motivation, and barriers. This discussion critically examines these aspects while identifying research gaps that require further exploration.

**Willingness to participate**

The willingness to participate in time bank is undeniably high across diverse populations, but this enthusiasm is not uniformly distributed and is heavily influenced by demographic and contextual factors. While younger generations show the highest levels of willingness, this trend raises questions about the inclusivity and accessibility of time bank systems for other demographic groups. For instance, [27] found that 82.67% of Chinese university students expressed interest in time bank, and [28] reported that 71% of Thai youth aged 18–23 were willing to participate. While these figures highlight the appeal of time bank among tech-savvy younger populations, they also underscore a potential bias in the design and promotion of these systems, which may inadvertently exclude older adults or those less comfortable with digital platforms. This raises a critical question: Are time bank systems truly inclusive, or do they primarily cater to younger, more techno-logically adept individuals?

Healthcare professionals, on the other hand, demonstrate moderate willingness, with 59.6% expressing interest in caregiving roles for older adults with disabilities [18]. While their altruistic motivations and professional alignment with caregiving are commendable, their participation is often constrained by professional obligations and a lack of supportive infrastructures. This suggests that time bank initiatives must address systemic barriers, such as time constraints and inad-equate support systems, to fully harness the potential of healthcare professionals. Without addressing these challenges, time bank risks becoming a supplementary rather than a transformative tool in community care.

Among older adults, willingness is shaped by factors such as physical quality of life, gender, and residence. [8] found that 58% of older Chinese adults were ready to engage in healthcare-related time bank, driven by their healthcare needs and demographic factors. However, this figure also implies that a significant portion of older adults remain hesitant, possibly due to mistrust, lack of awareness, or perceived complexity of the system. Interestingly, rural elderly populations show higher demand for mutual-support services [9], suggesting that time bank could play a pivotal role in addressing the unique needs of underserved communities. Yet, this potential remains largely untapped due to structural inefficiencies and limited outreach efforts.

Caregivers of dementia patients exhibit high willingness when enablers such as reduced caregiver burden, technologi-cal affinity, and professional support are present [15]. While this highlights the importance of situational and psychological factors in shaping participation, it also raises concerns about the sustainability of such engagement. Without ongoing support and tailored interventions, the initial enthusiasm of caregivers may wane, leading to disengagement and reduced effectiveness of time bank systems.

In conclusion, while willingness to participate in time bank is generally high, it is unevenly distributed and influenced by a complex interplay of demographic, contextual, and systemic factors. To realize the full potential of time bank, initiatives must move beyond one-size-fits-all approaches and adopt targeted strategies that address the unique barriers and motivators of different population groups. Only then can time bank become a truly inclusive and transformative tool for community engagement.

## Motivation factors

Motivational factors are undeniably central to driving participation in time bank, but their impact is not uniform and is often mediated by systemic and contextual challenges. These factors can be broadly categorized as intrinsic, extrinsic, and structural, each playing a distinct yet interconnected role in shaping engagement. Intrinsic motivations, such as altruism, self-promotion, and the desire to contribute to society, are frequently cited as significant drivers. For instance, [11] and [5] emphasize that engaging in meaningful work and experiencing unique social contributions are key motivators. Similarly, [21] and [22] highlight how the ideological principle of "equal time, equal value" fosters a sense of shared purpose among participants. However, while these intrinsic motivations are powerful, they are not sufficient on their own. Without addressing extrinsic and structural barriers, even the most altruistic individuals may find their enthusiasm waning over time.

Extrinsic motivators, such as tangible rewards like time coins and service hours, also play a critical role in influencing participation. [24] found that such rewards enhance engagement but cautioned that excessive demands could deter participants. This raises an important question: Are extrinsic rewards sustainable in the long term, or do they risk commodifying participation and undermining the intrinsic values that underpin time bank? Furthermore, the usability of time bank platforms is a pivotal extrinsic factor. [29] demonstrated that simplified, user-friendly systems significantly enhance participation, particularly among younger demographics. However, this also highlights a potential bias in design, as older adults or those with limited digital literacy may find such platforms inaccessible. If time bank systems are not designed with inclusivity in mind, they risk perpetuating existing inequalities rather than fostering community cohesion.

Social and organizational factors further complicate the motivational landscape. Strong social support networks, cross-sector collaboration, and co-production environments create enabling conditions for participation. [12] underscored the importance of organizational support mechanisms, such as regular meetings, incentives, and attentive coordination, in retaining participants. Reciprocity and trust in peer-to-peer exchanges also enhance commitment, as noted by [23]. Yet, these factors are not guaranteed and often depend on the availability of resources and institutional support. In resource-constrained settings, the absence of such structures can severely limit participation, particularly among marginalized groups.

Despite these motivational drivers, significant barriers persist. Usability challenges and a lack of awareness about time bank systems continue to hinder participation, as highlighted by [25]. These barriers are not merely technical but reflect deeper systemic issues, such as unequal access to resources and information. To truly enhance engagement, time bank initiatives must go beyond superficial fixes and address these structural inequities. Fostering value judgment, improving platform usability, and providing targeted support are essential steps, but they must be part of a broader strategy that prioritizes inclusivity and sustainability.

In conclusion, while motivational factors are crucial in driving participation in time bank, their effectiveness is contingent on addressing extrinsic and structural barriers. Without a holistic approach that balances intrinsic values with practical support, time bank risks becoming an exclusive rather than inclusive tool for community engagement. Future initiatives must critically evaluate these dynamics to ensure that time bank systems are accessible, equitable, and sustainable for all.

## Barriers to participation

Barriers to participation in time bank are not merely obstacles but systemic issues that reflect deeper inequities and inefficiencies within the model itself. These barriers span psychological, structural, social, and operational dimensions, each

compounding the challenges faced by potential participants. Psychological barriers, such as the negative impact of poor mental health, disproportionately affect older adults, reducing their willingness to engage [8]. Caregivers, for instance, face significant emotional strain, which not only limits their capacity to participate but also highlights the inadequacy of time bank systems in addressing the needs of those already burdened by caregiving responsibilities [18]. This raises a critical question: Can time bank truly support vulnerable populations if it fails to account for their psychological and emotional realities?

Structural barriers further exacerbate these challenges. Usability issues, logistical inefficiencies, and limited geographical accessibility are pervasive obstacles that undermine the practicality of time bank. [29] demonstrated that poorly designed platforms reduce behavioral intentions to participate, underscoring the urgent need for intuitive and accessible systems. Similarly, [3] identified that the hoarding of time credits and the lack of accessible locations to spend them diminish the functionality of time banks. These findings point to a fundamental flaw in the design of many time bank systems: they often prioritize theoretical ideals over practical usability, leaving participants frustrated and disengaged.

Social barriers, such as mistrust in the time bank concept and insufficient recognition of diverse skills, further limit participation. [26] and [13] highlighted that limited socialization opportunities can create a sense of isolation among participants, undermining the very social cohesion that time bank aims to foster. Moreover, ideological conflicts between participants with instrumental motivations (seeking practical benefits) and those with altruistic motivations (valuing community principles) often lead to dissatisfaction and disengagement [21]. This tension raises a provocative question: Is the principle of "equal time, equal value" inherently flawed, or does it simply require better implementation to accommodate diverse motivations?

Operational challenges, including resource-intensiveness, inefficiencies in the credit system, and a lack of government support, further hinder the scalability and sustainability of time bank systems [2,30]. These barriers are not merely logistical but reflect a broader lack of institutional commitment to alternative economic models. Without systemic interventions to address these inefficiencies and build trust among participants, time bank risks remaining a niche initiative rather than a transformative tool for community engagement.

In conclusion, the barriers to participation in time bank are not isolated issues but interconnected challenges that demand a holistic and systemic approach. Addressing these barriers requires more than superficial fixes; it necessitates a fundamental rethinking of how time bank systems are designed, implemented, and supported. Only by tackling these systemic inefficiencies can time bank fulfill its promise as an inclusive and sustainable model for community engagement.

## 5 Conclusion

Time bank extends beyond a simple system of service exchange; it represents a transformative approach to fostering social cohesion, reducing isolation, and promoting mutual aid. As a mechanism for strengthening communities, time bank offers significant potential; however, realizing its full benefits requires addressing key challenges and opportunities.

Existing research underscores the necessity of tailoring time bank initiatives to diverse demographic groups, each with distinct needs and motivations. Younger individuals may engage more effectively through gamified, technology-driven platforms, whereas older adults may benefit from simplified systems and health-related incentives. Additionally, caregivers require flexible participation options that accommodate their demanding schedules. Without targeted approaches, certain groups may remain underrepresented or excluded from time bank initiatives.

Sustained engagement in time bank is critical to its long-term viability. The factors influencing continued participation must be examined, including the role of social connections, tangible benefits, and a sense of purpose. A deeper understanding of these motivations is essential for developing strategies that ensure lasting involvement and commitment among participants.

Technological advancements present both opportunities and challenges for time bank implementation. While usability concerns remain a common barrier, emerging technologies such as artificial intelligence (AI) and blockchain offer potential

solutions. AI-driven systems could optimize service matching, while blockchain technology may enhance the security and transparency of time credit transactions. Further research is required to assess the feasibility and impact of these innovations on time bank systems.

Cultural and contextual variations significantly influence the effectiveness of time bank initiatives. While certain communities may successfully integrate time bank into their social fabric, others may encounter obstacles due to cultural norms, economic conditions, or societal values. A comprehensive understanding of these contextual factors is necessary to develop inclusive and adaptable models that cater to diverse populations.

Institutional and policy support play a crucial role in the scalability and sustainability of time bank. Without backing from governmental entities, nonprofit organizations, or cross-sector partnerships, time bank may struggle to overcome structural barriers. Supportive policies, public funding, and collaborative efforts across various sectors could enhance the credibility and reach of time bank systems.

Accurately measuring the impact of time bank remains a pressing challenge. Evaluating its effectiveness requires more than participant counts; it necessitates an analysis of broader social, economic, and psychological benefits. Key areas of assessment include improvements in mental health, community cohesion, and economic resilience. Establishing robust evaluation metrics will provide empirical evidence to support the expansion and adoption of time bank initiatives.

Ultimately, time bank serves as a powerful tool with the capacity to drive meaningful social change. However, its success depends on the effectiveness of its design, implementation, and ongoing support. Addressing existing research gaps and practical limitations will contribute to the development of inclusive, sustainable, and impactful time bank systems. The potential for transformative change exists, contingent upon continued efforts to refine and enhance this innovative approach.

## 6 Strengths and limitations

This scoping review offers a comprehensive overview of the current state of research on time bank, highlighting key factors such as willingness, motivators, and barriers to participation. One of its strengths lies in its broad scope, encompassing diverse populations, including younger individuals, older adults, healthcare professionals, and caregivers. By synthesizing findings from quantitative, qualitative, and mixed-methods studies, the review provides a nuanced understanding of the complexities surrounding time bank participation. Additionally, the identification of research gaps, such as the need for demographic-specific strategies, technological integration, and policy support, offers a clear roadmap for future research and practice. However, the review is not without limitations. First, restricting the review to English-language publications may have excluded valuable insights from non-English-speaking regions, limiting the global generalizability of the findings. Second, many of the included studies relied on self-reported data, which may be subject to recall bias and social desirability effects. Third, there remains a relatively small number of case studies across diverse geographic contexts, with much of the evidence concentrated in East Asia and Europe. Finally, while the review identifies gaps in the literature, it does not provide empirical solutions to these challenges. To address these limitations, future research should incorporate diverse linguistic and cultural contexts, conduct region-specific and in-depth case studies, and employ mixed-methods approaches to mitigate bias. Testing practical interventions to address identified barriers will also be critical to advancing the field and strengthening the role of time banking as an inclusive model for community engagement.

## Supporting information

S1 Fig. PRISMA-ScR Diagram. Generated at https://estech.shinyapps.io/prisma_fowdiagram/.
(DOCX)

S2 Fig. Publications per year (2010–2024).
(DOCX)

**S1 Table. Characteristics of the included studies (n = 30).**
(DOCX)

## Author contributions

**Conceptualization:** Che Chong Chin, Chong Mei Chan, Khadijah Alavi.

**Data curation:** Nur Shafira Azizan, Che Chong Chin.

**Formal analysis:** Nur Shafira Azizan, Che Chong Chin.

**Funding acquisition:** Tan Maw Pin.

**Investigation:** Nur Shafira Azizan.

**Methodology:** Nur Shafira Azizan, Che Chong Chin, Tan Maw Pin.

**Project administration:** Chong Mei Chan.

**Resources:** Pang Yong Kek, Khadijah Alavi.

**Supervision:** Che Chong Chin, Chong Mei Chan, Tan Maw Pin.

**Validation:** Che Chong Chin, Chong Mei Chan, Tan Maw Pin, Pang Yong Kek, Khadijah Alavi.

**Visualization:** Pang Yong Kek, Khadijah Alavi.

**Writing – original draft:** Nur Shafira Azizan.

**Writing – review & editing:** Nur Shafira Azizan, Chong Mei Chan.

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
