## [Decision Letter · Decision Letter 0]

10 Aug 2025

Dear Dr. Azizan,

We look forward to receiving your revised manuscript.

Kind regards,

Alhamzah F. Abbas, PhD

Academic Editor

PLOS ONE

Journal Requirements:

[We want to thank the Ministry of Higher Education for the Fundamental Research Grant Scheme (FRGS/1/2023/SS10/UM/02/5) awarded to Dr Che Chong Chin.].

[We want to thank the Ministry of Higher Education for the Fundamental Research Grant Scheme (FRGS/1/2023/SS10/UM/02/5) awarded to CCC.]

[We want to thank the Ministry of Higher Education for the Fundamental Research Grant Scheme (FRGS/1/2023/SS10/UM/02/5) awarded to Dr Che Chong Chin.]

Reviewers' comments:

Reviewer's Responses to Questions

**Comments to the Author**

1. Is the manuscript technically sound, and do the data support the conclusions?

Reviewer #1: Partly

Reviewer #2: Yes

2. Has the statistical analysis been performed appropriately and rigorously?

Reviewer #1: No

Reviewer #2: Yes

3. Have the authors made all data underlying the findings in their manuscript fully available?

Reviewer #1: Yes

Reviewer #2: Yes

4. Is the manuscript presented in an intelligible fashion and written in standard English?

Reviewer #1: Yes

Reviewer #2: Yes

Reviewer #1: Thank you for the opportunity to review “Willingness, Motivators and Barriers to Time Bank Participation: A Scoping Review”.

The article idea is quite interesting; however, it requires complete overhauling and require further improvements. Here are my suggestions and comment.

The abstract should exclude the headings such as objective, introduction, design etc.

How the study using Scooping review method addresses subjectivity bias of the authors is not discussed.

I believe Scooping review is booted with many citations and references, however, I cannot see that in the Introduction section and other parts.

The study lacks the rationale, objectives and research gap for conducting the Scooping review in the introduction section.

Line 107, statement about ethical approval should be removed. It is not something part of the manuscript purpose itself.

Developing questions in the methods section requires preliminary literature, which is lacking here to support the questions.

Why it leads to choose time frame January 2010 to December 2024, is not discussed.

Several databases were employed together, such as; electronic databases, which were MEDLINE, Embase, CINAHL, Cochrane Library, Google Scholar and Science Direct. relevant grey literature databases were also searched such as Grey Literature Report, OpenGrey, Web of Science Conference Proceedings, government documents, academic theses/dissertations) to locate and analyze relevant studies, reports, and conference abstracts for this review. The questions are several of these such as google Scholar suffers quality check. How the authors have addressed this shortcoming?

The authors have not focused on what actually PRISMA framework wants you to do. All page’s numbers mentioned in the PRISMA checklist seems irrelevant and lack of required information.

I have checked the key strings and there seems errors in it. Moreover, the use of key strings is not appropriate and this is one of the reasons the results are not clear and there may be critical articles missing. Same goes to other databases. I cannot see the use of Boolean operators and wild cards such as asterisk (*).

The authors can refer to these articles for clear understanding in addressing these comments;

Bibliometric analysis of finance and natural resources: past trend, current development, and future prospects.

Bibliometric analysis of global research trends on microfinance institutions and microfinance: Suggesting new research agendas. International Journal of Finance & Economics, 28(4), 3552-3573.

Research and Development Journey and Future Trends of Hollow Fiber Membranes for Purification Applications (1970–2020): A Bibliometric Analysis. Membranes, 11(8), 600.

I have not read the results due to the keywords and search strings used. I strongly believe, it can be further improved by addressing these comments, incorporating changes, and the appropriate use of the key strings to include relevant articles which may be missed due to this.

Reviewer #2: This article has a clear research objective and is well-organized. It systematically examines the scope and nature of evidence regarding willingness, motivations, and barriers to community participation in time bank systems, with a particular focus on older adults. The study identifies and discusses key factors that facilitate or hinder participation in time bank systems, effectively addressing gaps in the existing literature. This article demonstrates both innovation and practical relevance. However, the limited number of case studies across a wide geographic area restricts the scope of generalization and limits the ability to derive unique insights specific to certain regions.

.

Reviewer #1: No

Reviewer #2: No

---

## [Author Response · Author response to Decision Letter 1]

15 Sep 2025

Thank you for the helpful feedback. A detailed, point-by-point response to all reviewer and editor comments has been prepared and uploaded as a Word file. Please see the attached document for our full responses and corresponding changes made in the revised manuscript.

---

## [Decision Letter · Decision Letter 1]

11 Mar 2026

Willingness, Motivators and Barriers to Time Bank Participation: A Scoping Review

PONE-D-25-16688R1

Dear Dr. Azizan,

We’re pleased to inform you that your manuscript has been judged scientifically suitable for publication and will be formally accepted for publication once it meets all outstanding technical requirements.

Kind regards,

Kuo-Cherh Huang

Academic Editor

PLOS One

Additional Editor Comments (optional):

Reviewers' comments:

Reviewer's Responses to Questions

**Comments to the Author**

Reviewer #2: All comments have been addressed

2. Is the manuscript technically sound, and do the data support the conclusions?

Reviewer #2: Yes

3. Has the statistical analysis been performed appropriately and rigorously?

Reviewer #2: Yes

4. Have the authors made all data underlying the findings in their manuscript fully available?

Reviewer #2: Yes

5. Is the manuscript presented in an intelligible fashion and written in standard English?

Reviewer #2: Yes

Reviewer #2: Thank you for the opportunity to review this manuscript again. Overall, the topic of this article is academically valuable, the research methods are robust, and the results and analysis are clearly presented. The authors have thoroughly revised the manuscript and addressed the my comments. Therefore, I recommend this manuscript for publication.

.

Reviewer #2: No

---

## [Editor Report · Acceptance letter]

PONE-D-25-16688R1

PLOS One

Dear Dr. Azizan,

I'm pleased to inform you that your manuscript has been deemed suitable for publication in PLOS One. Congratulations! Your manuscript is now being handed over to our production team.

Kind regards,

on behalf of

Dr. Kuo-Cherh Huang

Academic Editor

PLOS One